# 17*β*-Estradiol Activates HSF1 via MAPK Signaling in ER*α*-Positive Breast Cancer Cells

**DOI:** 10.3390/cancers11101533

**Published:** 2019-10-11

**Authors:** Natalia Vydra, Patryk Janus, Agnieszka Toma-Jonik, Tomasz Stokowy, Katarzyna Mrowiec, Joanna Korfanty, Anna Długajczyk, Bartosz Wojtaś, Bartłomiej Gielniewski, Wiesława Widłak

**Affiliations:** 1Maria Sklodowska-Curie Institute – Oncology Center, Gliwice Branch, 44-101 Gliwice, Wybrzeże Armii Krajowej 15, Poland; patryk.janus@io.gliwice.pl (P.J.); agnieszka.toma-jonik@io.gliwice.pl (A.T.-J.); katarzyna.mrowiec@io.gliwice.pl (K.M.); joanna.korfanty@io.gliwice.pl (J.K.); a.dlugajczyk@ibb.waw.pl (A.D.); 2Department of Clinical Science, University of Bergen, Postboks 7800, 5020 Bergen, Norway; tomasz.stokowy@k2.uib.no; 3Laboratory of Molecular Neurobiology, Neurobiology Center, Nencki Institute of Experimental Biology, PAS, 3 Pasteur Street, 02-093 Warsaw, Poland; b.wojtas@nencki.gov.pl (B.W.); b.gielniewski@nencki.gov.pl (B.G.)

**Keywords:** bisphenol A, estrogen, Heat Shock Factor 1, MEK1/2, xenoestrogen

## Abstract

Heat Shock Factor 1 (HSF1) is a key regulator of gene expression during acute environmental stress that enables the cell survival, which is also involved in different cancer-related processes. A high level of HSF1 in estrogen receptor (ER)-positive breast cancer patients correlated with a worse prognosis. Here we demonstrated that 17*β*-estradiol (E2), as well as xenoestrogen bisphenol A and ER*α* agonist propyl pyrazole triol, led to HSF1 phosphorylation on S326 in ER*α* positive but not in ER*α*-negative mammary breast cancer cells. Furthermore, we showed that MAPK signaling (via MEK1/2) but not mTOR signaling was involved in E2/ER*α*-dependent activation of HSF1. E2­activated HSF1 was transcriptionally potent and several genes essential for breast cancer cells growth and/or ER*α* action, including *HSPB8*, *LHX4*, *PRKCE*, *WWC1*, and *GREB1*, were activated by E2 in a HSF1-dependent manner. Our findings suggest a hypothetical positive feedback loop between E2/ER*α* and HSF1 signaling, which may support the growth of estrogen-dependent tumors.

## 1. Introduction

Heat Shock Factor 1 (HSF1) is an evolutionarily conserved transcription factor, which is activated under stress conditions. Once activated, it regulates the expression of heat shock proteins (HSPs). HSPs function as molecular chaperones, which assist protein folding during synthesis and repair, or contribute to protein degradation under proteotoxic stress. Expression of HSPs is dependent not only on HSF1, as well as HSF1 function not being limited to the regulation of HSPs. HSF1 is also involved in different processes associated with development and growth, longevity, and fertility [1,2]. High HSF1 and HSPs expression were found in a broad range of tumors (and tumor cell lines), which usually correlates with a poor prognosis. HSF1 affects many aspects of cellular metabolism that are important for the cancer phenotype: it modulates signaling pathways associated with growth and proliferation, apoptosis, glucose metabolism, angiogenesis, and cell motility [3,4]. Additionally, it modulates signaling pathways altered by the expression of mutant oncogenic proteins, thus affecting the phenotype of cancer cells [5]. Such a phenomenon has been referred to in literature as “non-oncogenic addiction” [6]. A few reports point to the supportive role of HSF1 also in breast cancerogenesis. Especially, a high HSF1 level was associated with an increased mortality of estrogen receptor-positive (ER+) breast cancer patients [7,8].

In women, estrogens are produced naturally almost during the whole life. They are also used as part of some oral contraceptives and in estrogen replacement therapy for postmenopausal women. Additionally, many chemicals in the environment (i.e., bisphenol A) possess estrogenic features (so-called xenoestrogens), which may interfere with the functioning of the body and hormonal balance. Estrogens (especially the most potent physiological estrogen, 17*β*-estradiol, E2) are thought to be important in the pathogenesis of breast cancer. ER+ tumors represent up to 80% of all cases of breast cancers, which rely on supplies of estrogen to grow. Two different but complementary pathways have been proposed to contribute to the carcinogenicity of estrogen: E2 metabolites could bind to DNA and create depurinating DNA adducts leading to mutations, and/or E2 acting via the estrogen receptors could enhance proliferation of target mammary cells and increase the possibility of genomic mutations during DNA synthesis [9]. Two classes of estrogen receptor exist: (i) the intracellular ERs, which are members of the nuclear hormone family of intracellular receptors, and (ii) GPER1 (GPR30), which is a member of the rhodopsin-like family of the G protein-coupled and seven-transmembrane receptors. There are two different forms of the intracellular ERs (known as conventional or classical ERs), usually referred to as *α* and *β*, each encoded by a separate gene (ESR1 or ESR2, respectively). Ligand binding to ER leads to conformational changes that regulate the receptor activity, its interaction with other proteins and DNA [10]. In addition to the well-documented effects on transcription (genomic action), estrogen can activate signal transduction cascades outside of the nucleus (nongenomic action). This latter action could be mediated by GPER1 or a fraction of ERs localized at or near the cell membrane. Activated receptors initiate signaling cascades via secondary messengers that affect ion channels or increase nitric oxide levels in the cytoplasm, which ultimately leads to a rapid physiological response without involving gene regulation [11,12].

ERs in the inactive state are trapped in multimolecular chaperone complexes organized around the heat shock protein 90 (HSP90), containing p23 protein and immunophilin. Once bound E2, ER dissociates from the chaperone complexes and becomes competent to dimerize and to bind DNA [13]. Heat shock proteins, i.e., HSP27 oligomers, are also involved in the trafficking of ER to the cell membrane [14]. Thus, the activity of ERs is controlled by interactions with HSF1-regulated proteins. It could be anticipated that high HSF1 levels observed in breast cancer cells would support E2 action. On the other hand, however, though HSF1 could be activated by a plethora of factors, there is no data about the hypothetical impact of E2 on HSF1 activity. Therefore, we aimed to characterize the interplay between estrogen and HSF1 signaling in mammary epithelial tumorigenic and non-tumorigenic cells.

## 2. Results

### 2.1. 17β-Estradiol Induces HSF1 Phosphorylation on S326 in ERα-Positive Cells

We analyzed potential effects of E2 on activation of HSF1 using a range of established breast cell lines, i.e., two non-tumorigenic (MCF10a and MCF12a) and four originated from cancer patients (MCF7, SKBR3, MDA-MB-468, BT-547) with a different ER status (Figure 1A). These cells were treated with 10 nM E2 for 1 or 2 hours and then the phosphorylation of HSF1 on S326 (which plays a critical role in the induction of the transcriptional competence [15]) and S303+307 (which contribute to down-modulation of HSF1 activity [16]) were analyzed (Figure 1B). We did not observe any changes in S303+307 phosphorylation after E2 treatment, yet an elevation of HSF1 phosphorylation on S326 was noted in ER*α*-positive MCF7 cells (ER*α*, ER*β*, and GPR30-positive). However, there were no changes in S326 phosphorylation in ER*α*-negative cell lines, although some of them (e.g., SKBR3) were ER*β* or GPR30-positive. This observation suggests that signaling leading to S326 HSF1 phosphorylation could be dependent on ER*α* but neither ER*β* nor GPR30.

To confirm that ER*α* is indispensable for E2-induced phosphorylation of HSF1 we treated MCF7 cells with selective ER*α* agonist (propyl pyrazole triol (PPT)) or xenoestrogen bisphenol A (BPA) which both induced ER*α* phosphorylation on S118 (the major site phosphorylated in response to estradiol [17]) in a similar way as E2 (Figure 2A). We found that PPT and BPA were able to increase HSF1 phosphorylation on S326. On the contrary, HSF1 phosphorylation on S326 was not markedly induced after treatment with antagonists of ER*α*, i.e., 4-OHT or ICI 182,780 (fulvestrant) (Figure 2B). Interestingly, ER*α* antagonists induced transient phosphorylation of ER*α* on S118, although much weaker than induced by E2. For further experiments, before E2 treatment we pretreated cells with 4-OHT and ICI 182,780 in conditions which did not cause ER*α* phosphorylation. We showed that E2 action on HSF1 and ERα was abrogated by antagonists of ER*α* (Figure 2C). To confirm the necessity of ER*α* expression for E2-induced phosphorylation of HSF1 we also downregulated ER*α* in MCF7 cells using specific siRNA. ER*α* silencing resulted in the lack of HSF1 phosphorylation on S326 after E2 treatment (Figure 2D). Furthermore, we observed the E2-stimulated increase in HSF1 phosphorylation on S326 in other ER*α*-positive T47D cells (Appendix A), though the level of ER*α* expression was much lower in T47D than in MCF7 cells. Hence, these data collectively confirmed the importance of ER*α* in E2-induced activation of HSF1.

### 2.2. MEK/ERK Signaling is Involved in E2-Induced Phosphorylation of HSF1 on S326

Looking for kinases which could be involved in HSF1 phosphorylation after E2 treatment of MCF7 cells, we first utilized proteome profiler Human Phospho-Kinase Array. We searched for kinases activated both by E2 and heat shock (a typical treatment which triggers HSF1 activation). Using these conditions we detected enhanced (at least twice above control untreated cells) phosphorylation of ERK1/2 (on T202/Y204, T185/Y187), JNK1/2/3 (on T183/Y185, T221/Y223), and CREB (on S133) (Appendix A), which indicated that MAPK signaling could be involved in HSF1 phosphorylation after both treatments. Earlier reports have shown that HSF1 could directly be phosphorylated on S326 by all members of the p38/MAPK family [18], AKT [19], and mTOR [20]. Thus, we further studied the time-course of HSF1 phosphorylation on S326 after E2 treatment with respect to phosphorylation of ERK1/2 as well as AKT, mTOR and p70S6K (the downstream substrate of mTOR that also acts in the PI3K/AKT pathway). E2 induced phosphorylation of ER*α* on S118 already after 15 minutes of treatment (with the higher level after 30–60 minutes), while HSF1 phosphorylation took place most effectively within 1–4 hours during E2 treatment (Figure 3A). It is noteworthy that the increased level of HSF1 phosphorylation coincided with the enhanced phosphorylation of ERK1/2 on T202/Y204. On the other hand, AKT phosphorylation on S473 and mTOR on S2448 were not affected by E2 treatment (although the level of phosphorylated p70S6K was transiently increased).

To confirm the involvement of ERK1/2 signaling in HSF1 phosphorylation, we pre-treated MCF7 cells with U0126 (a highly selective inhibitor of MEK1/2, upstream kinases which activate ERKs) before the addition of E2. The E2-induced phosphorylation of ERK1/2 (T202/Y204) was completely blocked by U0126. This was associated with abrogated phosphorylation of HSF1 (S326), as well as partially reduced phosphorylation of ER*α* (S118) and p70S6K (T389) but elevated phosphorylation of AKT (S473). Although we did not notice any changes in AKT phosphorylation under E2 treatment, we decided to pretreat MCF7 cells with LY294002 (an inhibitor of PI3K, which is a part of PI3K/AKT/mTOR signaling). This caused a decrease in AKT (S473), mTOR (S2448), and p70S6K (T389) phosphorylation, but there were no changes in E2-induced phosphorylation of HSF1 or ERK1/2. Also, pretreatment of MCF7 cells with rapamycin (an inhibitor of mTORC1) did not lead to a decrease in phosphorylation of HSF1 induced by E2 (Figure 3B). To summarize, we stated that inhibition of signaling via MEK1/2 and ERK1/2, but neither PI3K/AKT nor mTORC1 could abrogate phosphorylation of HSF1 induced by E2.

### 2.3. HSF1 Phosphorylated on S326 under E2 Treatment is Transcriptionally Active

E2-induced phosphorylation of HSF1 on S326 (about 2-fold) seems to be moderate in comparison to strong induction by heat shock (about 10-fold, not shown). Thus, we analyzed whether HSF1 is transcriptionally active in response to E2 treatment. First, we analyzed the expression of several *HSP* genes containing classical heat shock elements (HSEs) in their promoters, typical targets of HSF1 (Figure 4A). We found up-regulation of *HSPH1* and *HSPB8* transcription, but not *HSPA1*, in MCF7 cells treated with E2 (Figure 4B). Consistently, we found that binding of HSF1 in promoters of *HSPH1* and *HSPB8* genes, but not *HSPA1* gene, was markedly enhanced after E2 treatment (Figure 4C).

To find other molecular targets of E2-activated HSF1, we performed high-throughput analysis for the global identification of HSF1 binding sites (ChIP-Seq) and global expression profiles (RNA-Seq). There were 255 HSF1 binding sites in the whole genome showing increased occupancy after E2 treatment, of which 144 were located in the regulatory region of 138 genes (the putative regulatory region for each gene was arbitrarily defined as a region spanning from −3000 base pairs upstream of the transcription start site to the end of the coding sequence) (Appendix A). In comparison, there were over 7000 genes that showed increased HSF1 binding after heat shock (HS) in their putative regulatory regions (118 genes were common for both treatments). RNA-Seq analysis revealed that among 138 genes with E2-enhanced binding of HSF1 in putative regulatory regions there were only 28 genes that showed expression changes after E2 treatment: 23 upregulated (among them *GREB1* gene, an early response gene in the estrogen receptor-regulated pathway) and 5 downregulated (Figure 5A; see also Appendix A). Gene ontology analysis revealed that only two pathways were enriched in this gene set: CCKR signaling map and Wnt signaling pathway, both belonging to the highly enriched pathways in the E2-regulated gene set identified by RNA-Seq (Figure 5B). Detailed sequence analysis of ChIP-Seq detected peaks revealed the presence of HSE motifs in regulatory regions of 17 such genes (Figure 5C; see Appendix A for HSF1 peak localization, motif sequence, and E2-induced expression level for these genes). Nine genes (for which it was possible to design PCR primers covering detected HSE) further validated qPCR. Although expression of all nine genes was upregulated after E2 treatment, E2-induced HSF1 binding to HSE motifs was confirmed only in the case of four genes: *HSPB8*, *LHX4*, *PRKCE*, *WWC1* (Figure 5D), which suggested direct regulation of these genes by HSF1. In parallel, a similar analysis was carried out for *GREB1* gene, where the classic HSE motif was not found in the ChIP-Seq detected peak, but this gene was interesting because of its participation in the regulation of the estrogen receptor pathway. We confirmed the E2-induced upregulation of this gene, especially of isoform c (NM_148903), as well as enhanced HSF1 binding in the promoter region of this transcript variant. It should be noted, however, that for the majority of validated genes (except *GREB1* and *HSPB8*) HSF1 enrichment was significantly higher after HS treatment than after E2 stimulation (data not shown), which was consistent with a higher HSF1 phosphorylation level after HS.

## 3. Discussion

Looking for the cross-talk between estrogen and HSF1 signaling pathways we found that E2 can activate HSF1 in ER*α*-positive breast cancer cells through MEK1/2 signaling. Furthermore, we demonstrated that E2-activated HSF1 was transcriptionally potent and was able to bind to the regulatory regions of some genes and modulate their expression. This is a very important finding bearing in mind that up to 80% of all breast cancer cases rely on supplies of the estrogen to grow, while HSF1 is frequently overexpressed in breast cancer and its high level in ER-positive cases negatively correlates with the survival time of patients [7,8]. It has been evident that HSF1-regulated chaperones are indispensable for estrogen receptors signaling. Here we have revealed for the first time that E2 (as well as other ER*α* agonists, e.g., xenoestrogens) can enhance HSF1 action. A hypothetical positive feedback loop involving E2/ER*α* and HSF1/HSPs is therefore possible. The cross-talk between estrogen and HSF1 signaling pathways may be important in other ER*α*-positive cells as well. In fact, it was observed that high HSF1 expression in endometrial carcinoma was significantly associated with aggressive disease and poor survival, also among ER*α*-positive patients presumed to have a good prognosis [21].

Phosphorylation on S326 is crucial for full activation of HSF1 in stress conditions [15,22]. Here we observed a modest HSF1 phosphorylation on S326 in response to E2 when compared to its hyperphosphorylation in response to heat shock. Nevertheless, HSF1 acquired the ability to bind to HSE motifs and to regulate the expression of some genes in E2-treated cells. Thus, we confirmed that HSF1 hyperphosphorylation is not obligatory to achieve transcriptional competence by HSF1 and depending on stimuli HSF1 can realize a distinct genome-wide transcriptional program [23,24]. We found that HSF1 was activated in response to E2 only in MCF7 and T47D cells (expressing all three estrogen receptors: ER*α*, ER*β* and GPR30) and we proved that ER*α* is critical for this signaling. It is known that ER*α* is palmitoylated on the Cys447 residue which leads to its translocation to the membrane and dimerization within seconds of E2 exposure. ER*α* redistribution allows its association with adaptors and/or signaling proteins, among others tyrosine kinase Src [12,25]. This contributes to the activation of MAPK (i.e., RAS/MEK/ERK) and PI3K/AKT signaling cascades [26,27]. We found that ERK1/2 signaling was activated in E2-treated MCF7 cells, while the PI3K/AKT pathway was only activated when the ERK1/2 pathway was blocked. Thus, we postulate that a membrane-associated ER signaling pathway via activation of MAPK, but not PI3K/AKT, is involved in E2-induced phosphorylation of HSF1. Moreover, we postulate the role of MEK1/2 in this mechanism. It was previously shown that MEK, rather than ERK, physically interacts with and activates HSF1 [28]. Furthermore, MEK2 (MAP2K2; dual specificity mitogen-activated protein kinase 2) was identified among 40 phosphoproteins specifically modified in MCF7 cells exposed to E2 [29]. Earlier reports showed that HSF1 could be phosphorylated on Ser326 by all members of the p38/MAPK family [18]. Alternatively, it was shown that HSF1 may be phosphorylated on S326 by mTORC1 [20], which can be a downstream target of signaling via RAS/RAF/MEK/ERK as well as via RAS/PI3K/AKT/mTOR pathways. However, we observed no increase in mTOR phosphorylation on S2448 under E2 treatment (yet p70S6K, a downstream target of mTOR was phosphorylated; [30] and current data). Moreover, inhibition of mTOR by rapamycin treatment was not able to prevent HSF1 phosphorylation, which suggests that mTOR signaling is not involved in E2-induced phosphorylation of HSF1.

We have found that up-regulation of several genes, including *HSPB8*, *LHX4*, *PRKCE*, *WWC1*, and *GREB1*, could be HSF1-dependent in E2-treated MCF7 cells. Generally, these genes code for proteins important for estrogen signaling and breast cancer development. HSPB8 (heat shock protein beta-8) has been shown to be induced by E2 [31] and to modulate E2-induced proliferation and migration of MCF7 cells [32]. Accordingly, *HSPB8* was identified as a candidate tumor progression gene [33]. Also, *WWC1*, *LHX4*, and *GREB1* have been found among E2-induced genes in MCF7 cells [34]. *WWC1* (WW and C2 domain containing 1) encodes ER*α*-interacting protein, which participates in ER*α* transactivation in breast cancer cells [35]. *GREB1* (growth regulating estrogen receptor binding 1) is an early estrogen-responsive gene [36], and its expression correlates with estrogen levels in breast cancer patients [37]. Only *PRKCE* (protein kinase C epsilon) has not been linked to estrogen action in breast cancer yet, although this protein has been shown to enhance the survival, anchorage-independent growth, and lung metastasis of the LM3 murine mammary tumor cell line (which lacks PR and ER expression) [38].

## 4. Materials and Methods

### 4.1. Cell Lines

Human non-tumorigenic MCF10a and MCF12a cell lines and human breast cancer cell lines (MCF7, SKBR3, BT-549, MDA-MB-468, and T47D) were used. MCF10a, MCF12a, MCF7, SKBR3, and MDA-MB-468 cell lines were provided and/or authenticated by the American Type Culture Collection (ATCC), the T47D cell line—by the European Collection of Authenticated Cell Cultures (ECACC). MCF10a and MCF12a cells were cultured in DMEM/F12 medium supplemented with 5% horse serum, 5 µg/mL insulin (Sigma-Aldrich, St. Louis, MO, USA), 0.5 µg/mL hydrocortisone (Sigma-Aldrich), 20 ng/mL epidermal growth factor (Sigma-Aldrich) and cholera toxin (Sigma-Aldrich). MCF7 cells were cultured in DMEM/F12 medium supplemented with 10% fetal bovine serum (FBS). SKBR3, BT549, and MDA-MB-468 cells were cultured in RPMI medium supplemented with 10% FBS. T47D cells were cultured in high glucose DMEM supplemented with 10% FBS and 5 μg/mL insulin (Sigma-Aldrich).

### 4.2. Chemicals

The following chemicals were purchased: 17*β*-estradiol (E2), 4’-hydroxytamoxifen (4-OHT), ICI 182,780, 4,4’,4’’-(propyl-c-pyrazole-1,3,5-triyl)-tris-phenol (PPT), bisphenol A (BPA) from Sigma-Aldrich; rapamycin from Calbiochem-Merck; U0126 and LY294002 from Cell Signaling Technology.

### 4.3. Treatments

Cells were seeded on plates. The next day the medium was replaced into phenol-free DMEM/F12 medium supplemented with 5% dextran-activated charcoal-stripped FBS and 48 hours later cells were subjected to hyperthermia at 43 °C (15 minutes for ChIP/ChIP-Seq or one hour for RT-PCR/RNA-Seq) or 17*β*-estradiol was added to final concentration of 10 nM (or 1 nM and 100 nM; an equal volume of ethanol was added as vehicle control) for indicated time (1 h and 2 h for ChIP/ChIP-Seq; 2 h and 4 h for RT-PCR/RNA-Seq; for western blotting see details in the descriptions of the Figures). The same scheme was used for cell treatment with PPT, BPA, 4’OHT or ICI 182,780. U0126 (10 µM; MEK1 and MEK2 inhibitor), LY294002 (25 µM; PI3Ks inhibitor) or rapamycin (40 µM; mTOR inhibitor) were added two hours before E2 treatment.

### 4.4. Western Blotting

Whole protein extracts were prepared using RIPA buffer consisting of 50 mg sodium dodecyl cholate (POCh, Gliwice, Poland), 0.1% sodium dodecyl sulfate (Sigma-Aldrich), 10% Nonidet-40 (Fluka), 1 mM phenylmethylsulfonyl fluoride (PMSF, Sigma-Aldrich), protease inhibitor mixture CompleteTM (Roche) and phosphatase inhibitors PhosStopTM (Roche) dissolved in PBS. Total protein content was determined using the Protein Assay kit (Bio-Rad; Hercules, CA, USA). Proteins (25 µg) were separated on 10% SDS-PAGE gels and blotted to 0.45-μm pore nitrocellulose filter (GE Healthcare) using Trans Blot Turbo system (Bio-Rad) for 10 min. The filter was blocked for 60 min in 5% non-fat milk in TTBS (250mM Tris-HCl pH 7.5, 0.1% Tween-20, 150mM NaCl) and next incubated at 4 °C overnight with a primary antibody (Appendix A). The primary antibody was detected by an appropriate secondary antibody conjugated with horseradish peroxidase and visualized by enhanced chemiluminescence (ECL) kit (Thermo Scientific). Control of protein loading was done using an antibody against ACTB conjugated with horseradish peroxidase or HSPA8 (Appendix A). The experiments were repeated at least in triplicate and blots were subjected to densitometric analyses using ImageJ software to calculate relative expression alongside loading controls (statistical significance was calculated using *T*-test).

### 4.5. ERα Downregulation

MCF7 cells were seeded on 3-cm plates at the density of 15 × 10^4^ cells in DMEM/F12 medium supplemented with 10% FBS. The next day the medium was replaced with phenol-red free DMEM/F12 medium supplemented with 5% dextran-activated charcoal-stripped FBS and cells were transfected with either negative control siRNA or ER*α*-targeting siRNAs (Dharmacon, ON-TARGETplus pools, #D-001810-10-20 or #L-003401-00-0020, respectively) using DharmaFECT^TM^ transfection reagent (Dharmacon) according to supplier procedure. Forty-eight hours later E2 was added to a final concentration of 10 nM. Cells were incubated for one or two hours and collected for western blot.

### 4.6. RNA Isolation, cDNA Synthesis, and RT-qPCR

Total RNA was isolated using the Direct-ZolTM RNA MiniPrep Kit (Zymo Research), digested with DNase I (Worthington Biochemical Corporation, Lakewood, NY, USA), converted into cDNA and used for RT-qPCR. Quantitative PCR was performed using a BioRad C1000 TouchTM thermocycler connected to the head CFX-96. Each reaction was performed at least in triplicate and contained: 1 × PCR Master Mix SYBRGreen (A&A Biotechnology, Gdynia, Poland), 200 nmoles of each primer, and cDNA template (the equivalent of 10 ng of transcribed RNA). The set of delta-Cq replicates for control and test samples were normalized according to the geometric mean of the reference *GAPDH* and *HNRNPK* housekeeping genes. Normalized values were used for expression fold change calculation using the double delta Cq method (by determining the median, maximum and minimum value) and for estimation of the *p* values. The primers used in these assays are described in Appendix A.

### 4.7. Global Gene Expression Profiling

Total RNA was isolated using the Direct-ZolTM RNA MiniPrep Kit (Zymo Research) and digested with DNase I (Worthington Biochemical Corporation). Preparation of cDNA libraries and sequencing by Illumina HighSeq 2500 (run type: single read, read length: 1 × 50 bp) were carried out by GATC Biotech. Raw RNA-Seq reads were aligned to human genome hg19 using tophat2 [39] with Ensembl genes transcriptome reference. Aligned files were processed using Samtools [40]. Furthermore, reads aligned in the coding regions of the genome were counted using FeatureCounts [41]. Finally, read counts were normalized using DESeq2 [42], then normalized expression values were subject to differential analysis (mean based fold change) in the R/Bioconductor programming environment. Assuming multiple filtering and planned qPCR validation of candidate genes, a moderate significance threshold of differences was applied: changes (treatment versus control) were considered significant if signal ratios were >1.5 or <0.67. The raw RNA-Seq data were deposited in the NCBI GEO database; acc. no. GSE137559.

### 4.8. Chromatin Immunoprecipitation and ChIP-qPCR

The ChIP assay was performed according to the protocol from the iDeal ChIP-Seq Kit for Transcription Factors (Diagenode). Cells were fixed using 1% formaldehyde in PBS for 10 min at room temperature. Fixation was quenched by glycine (125 mM final concentration) and nuclei were isolated. Chromatin was sheared using Bioruptor^®^ PLUS combined with the Bioruptor^®^ Water cooler & Single Cycle Valve (at HIGH power setting) with 17 cycles of 30 sec shearing followed by 30 sec of standby and chromatin fragments with approximate length 100–600 bp were obtained. For each IP reaction, 30 μg of chromatin and 4 μg of rabbit anti-HSF1 polyclonal antibody (ADI-SPA-901, Enzo) was used. For negative controls, chromatin samples were processed without antibody (mock-IP). Obtained DNA fragments were used for global profiling of chromatin binding sites or gene-specific ChIP-qPCR analysis using specific primers covering the known heat shock elements (HSE). The set of delta-Cq replicates (difference of Cq values for each ChIP-ed sample and corresponding input DNA) for control and test sample were used for HSF1 binding calculation (as a percent of input DNA) and estimation of the *P* values. HSE motifs in individual peaks were identified using MAST software from the MEME Suite package [43]. Sequences of used primers are presented in Appendix A.

### 4.9. Global Profiling of Chromatin Binding Sites

In each experimental point, six ChIP biological replicates (each from 30 μg of input chromatin) were collected and combined in one sample before DNA sequencing. Immunoprecipitated DNA fragments and input DNA were sequenced using the HiSeq 1500 system with TruSeq workflow (Illumina). Raw sequencing reads were analyzed according to standards of ChIP-Seq data analysis as described below. Quality control of reads was performed with FastQC software (www.bioinformatics.babraham.ac.uk/projects/fastqc) and low-quality sequences (average phred <30) were filtered out. Remained reads were aligned to the reference human genome sequence (hg19) using the Bowtie2.0.4 program [44]. Peak detection was carried with the MACS program [45], whereas the outcome was annotated with Homer package [46]. Peak intersections and their genomic coordinates were found using Bedtools software [47]. The input DNA was used as a reference because no sequences were obtained using a mock-IP probe. The significance of differences between control untreated cells and cells subjected to hyperthermia or 17*β*-estradiol was estimated using MACS software. All ChIP-Seq-detected differential peaks were filtered and only peaks with a high score (*P* value <10^−6^, number of tags >40) were taken for counting analysis. The raw ChIP-Seq data were deposited in the NCBI GEO database; acc. no. GSE137558.

### 4.10. GO Enrichment Analyses

Enrichment analyses on gene sets were performed using the PANTHER classification system version 14.1 [48]. Genes were classified according to PANTHER pathways annotation.

## 5. Conclusions

We documented that estrogen (as well as xenoestrogens) through ER*α* and MAPK can activate HSF1 in breast cancer cells. Furthermore, HSF1 can modulate the expression of genes involved in E2-stimulated signaling. The presented results suggest that HSF1 activated by estrogen could support cancer cells’ growth independently of its cytoprotective function mediated by chaperones. This explains why patients with ER-positive breast cancer overexpressing HSF1 have a worse prognosis.

## Figures and Tables

**Figure 1 cancers-11-01533-f001:**
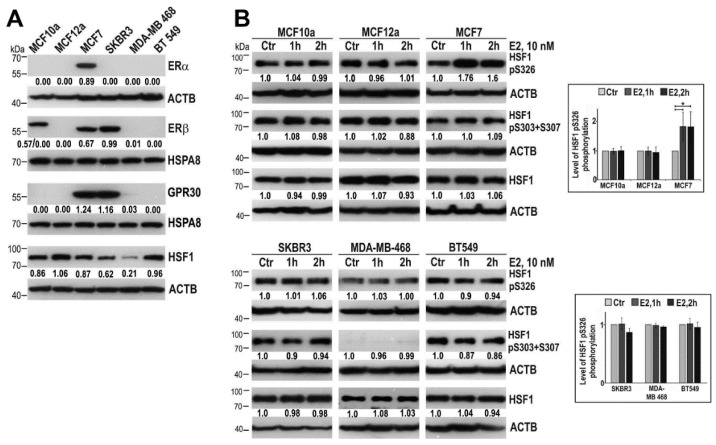
17*β*-estradiol (E2) induces Heat Shock Factor 1 (HSF1) phosphorylation on S326 in ERs/GPR30-positive MCF7 cells but not in ER*α*-negative mammary cells. (**A**) Expression of estrogen receptors, ER*α*, ER*β*, and GPR30, as well as HSF1 in human mammary non-tumorigenic (MCF10a and MCF12a) and tumorigenic (MCF7, SKBR3, MDA-MB-468, and BT549) cells. (**B**) The above cell lines were treated with 10 nM E2 for one hour or two hours and the whole-cell extracts were analyzed by western blot. The right panel shows the results of densitometric analyses. An asterisk indicates a significant difference (*p* < 0.05) from the control (Ctr) value. ACTB and HSPA8 were used as controls for protein loading. Numbers below blots represent protein bands’ intensity ratios to adequate controls after normalization against ACTB.

**Figure 2 cancers-11-01533-f002:**
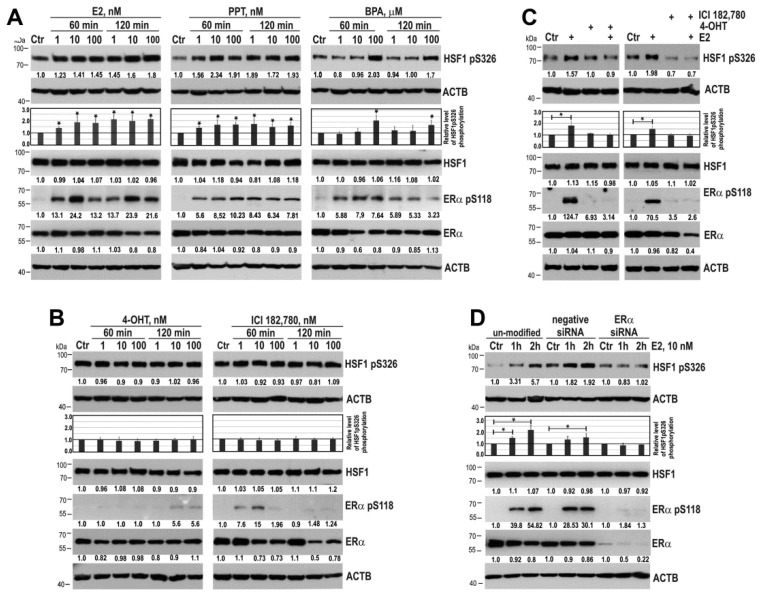
ER*α* is indispensable for HSF1 phosphorylation on S326 in MCF7 cells. (**A**) An influence of ER*α* agonists (E2, PPT, BPA) and (**B**) E2 antagonists (4-OHT and ICI 182,780) on HSF1 phosphorylation. (**C**) Effect of E2 on HSF1 phosphorylation after pretreatment of cells with 4-OHT or ICI 182,780 (added at concentration 100 nM for one hour before treatment with 10 nM E2 for one hour or two hours, respectively). (**D**) Effect of E2 on HSF1 phosphorylation in cells with down-regulated ER*α*. ER*α* phosphorylation on S118 was used as a marker of ER*α* activation, and ACTB was used as a protein loading control. The results of densitometric analyses are shown in the charts. An asterisk indicates a significant difference (*p* < 0.05) from the control (Ctr) value. Numbers below blots represent protein bands’ intensity ratios to adequate controls after normalization against ACTB.

**Figure 3 cancers-11-01533-f003:**
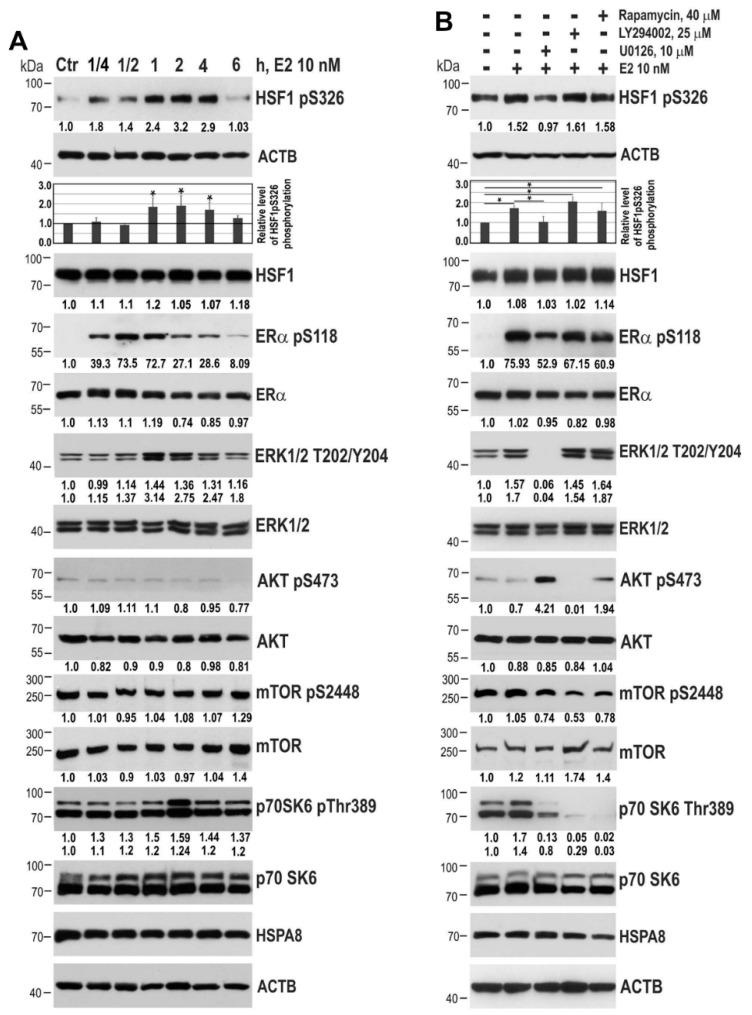
ERK1/2 and mTOR are involved in E2-induced HSF1 phosphorylation in MCF7 cells. Western blot analyses: (**A**) The time-course analysis of HSF1 phosphorylation on S326, ER*α* on S118, ERK1/2 on T202/Y204, AKT on S473, mTOR on S2448 and p70S6K on T389 in cells treated with 10 nM E2. (**B**) HSF1 phosphorylation on S326 in cells treated with E2 for one hour alone or pretreated with 10 µM U0126 (MEK1/2/ERK1/2 signaling inhibitor), 25 µM LY294002 (PI3K/AKT signaling inhibitor) or 40 µM rapamycin (mTOR inhibitor) for two hours. ACTB and HSPA8 were used as controls for protein loading. The results of densitometric analyses are shown in the charts. An asterisk indicates a significant difference (*p* < 0.05) from the control (Ctr) value. Numbers below blots represent protein bands’ intensity ratios to adequate controls after normalization against ACTB.

**Figure 4 cancers-11-01533-f004:**
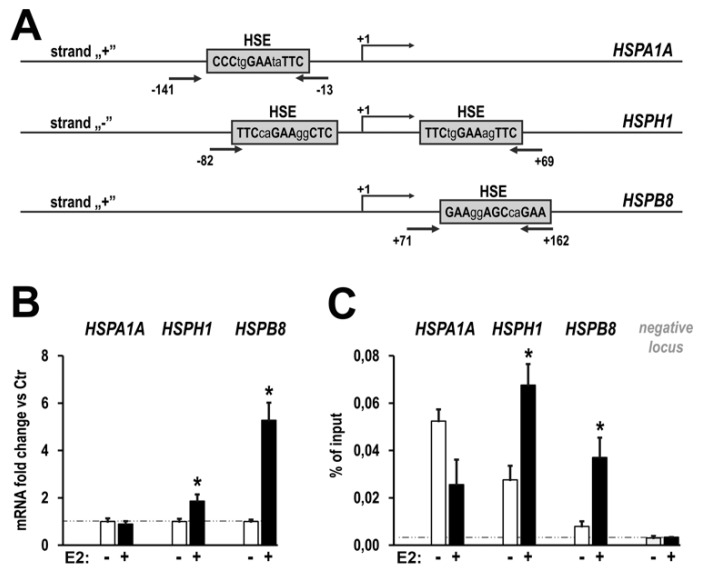
E2-induced HSF1 is transcriptionally active. (**A**) The architecture of promoter regions of three selected HSP genes: location of the heat shock elements (HSEs) and regions amplified in ChIP-qPCR analysis (arrows) are marked. (**B**) qPCR analysis of mRNA levels of selected *HSP* genes in response to E2 (10 nM, four hours). (**C**) qPCR analysis of E2-induced (10nM, one hour) HSF1 binding to the known HSE sequences in promoter regions of selected genes and to the control region (negative locus). Asterisks indicate the statistical significance of differences (*p* < 0.05) between control and test samples.

**Figure 5 cancers-11-01533-f005:**
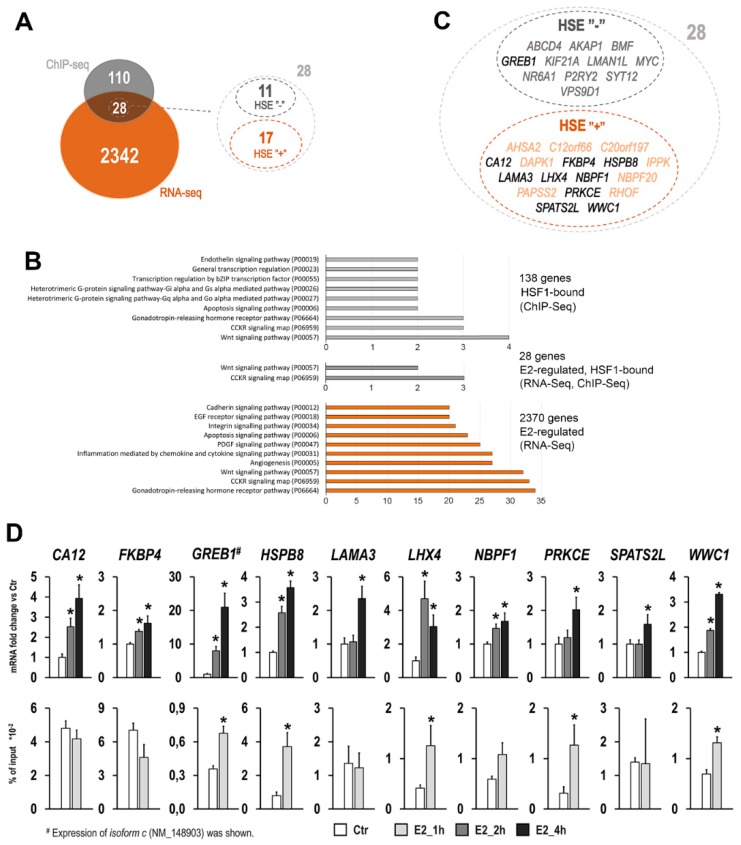
Identification of genes affected by E2-treatment potentially regulated by HSF1 in MCF7 cells. (**A**) A number of genes with E2-induced binding of HSF1 in their regulatory region (identified by ChIP-Seq) and/or altered expression profile (identified by RNA-Seq). To find HSE motifs and to identify genes directly regulated by HSF1, the detailed analysis within the ChIP-Seq-detected peaks was performed. (**B**) Gene ontology analyses showing numbers of genes in pathways from the PANTHER classification system. In the case of gene sets selected in ChIP-Seq (138 or 28 genes), only pathways with more than two genes are shown. In the case of all E2-regulated genes, only ten pathways with the highest number of genes are shown. (**C**) List of 28 E2-regulated genes with and without HSE motifs in their regulatory regions. Bolded symbols indicate the genes selected for further validation experiments. (**D**) Validation analysis for ten selected genes. Gene expression (upper row) and binding of HSF1 to detected HSE sequences in regulatory regions (bottom row) were analyzed by qPCR. Asterisks indicate the statistical significance of differences (*p* < 0.05) between control and test samples.

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
