# Peer review of "17β-Estradiol Activates HSF1 via MAPK Signaling in ERα-Positive Breast Cancer Cells"

_cancers, 2019, doi:10.3390/cancers11101533_

Round 1

Reviewer 1 Report

The manuscript by Vydra et al., explores the effect of estrogen receptor (ER) signaling on heat shock factor 1 (HSF1). Although a high HSF1 level was associated with an increased mortality of ER+ breast cancer patients, there was no data about the impact of ER signaling on HSF1 activity. In their study, the authors identified that ER signaling activates HSF1 via MAPK. Overall, this is a nice body of work with convincing data. The authors did well to examine the effect of ER signaling on HSF1 using biochemical approaches and high-throughput analyses with CHIP-seq and RNA-seq. However, the authors must address issues below before this manuscript can be considered for publication:

In some figures (e.g. Fig 1B), protein labels are placed on the left side of blots. However, in others (e.g. Fig 1A), protein labels are placed on the right side. To be consistent, the authors should place protein labels on the left side and include molecular weight marks on the right side for all figures.

In Fig 1B, E2 treatment resulted in decreased levels of HSF1 in ER-negative MDA-MB-468 cells. However, this interesting observation was not mentioned in the manuscript. While levels of HSF1 p326 did not change in MDA-MB-468 cells, one may argue that HSF1 phosphorylation level should be examined in relation to its total level (HSF1 p326/total HSF1). Under such circumstance, HSF1 phosphorylation would increase in MDA-MB-468 cells. Also of note, another ER- triple negative cell line used in the study, BT549, did not yield the same effect.

Graphs shown in figures 2 and 3 show positive values of y-axes pointing downward. This can be confusing to readers so it is recommended that the authors invert these graphs. Doing so would also make them to be consistent with graphs in figures 1, 4 and 5.

The authors should present lists of genes from CHIP-seq and RNA-seq analyses as part of the supplementary data. In addition, the authors should perform gene ontology analyses of these genes using tools such as DAVID (https://david.ncifcrf.gov/) and present the finding. Doing so would help readers to better appreciate where HSF1 signaling stands among different pathways/processes that are altered upon E2 induction.

For genes shown in Fig 5C, the authors should verify that MAPK signaling is required by examining their levels in the presence of the MAPK inhibitor, U0126. Also the authors found Erk, JNK and CREB to be phosphorylated by E2 using the phospho-kinase array. Since JNK is also known to phosphorylate HSF1 (Dai et al. J Biol Chem 2000, Park & Liu, J Cell Biochem 2001), JNK may play part in activating genes listed in Fig 5C. Therefore, the authors should examine gene expression in the presence of JNK inhibitors such as SP600125. In addition, examining protein levels using western blots would nicely complement findings in Fig 5C.

Reviewer 2 Report

HSF1 is a heat shock factor that is regulated by stress. Reactive oxygen species are important stress generators in the cell. 17β-Estradiol was shown to induce ROS, further ROS were extensively reported to activate MAPK pathway. It will be worth to check if the ROS produced by 17β-Estradiol is activating MAPK that is further leading to Phosphorylation of HSF1. Authors can add antioxidants to block ROS and verify the results. This experiments will strengthen the mechanism of paper by elucidating the upstream activators of MAP kinases, which are Phosphorylating HSF1.
